# Scanning tunnelling spectroscopy as a probe of multi-**Q** magnetic states of itinerant magnets

Maria N. Gastiasoro[1], Ilya Eremin[2,3], Rafael M. Fernandes[4] & Brian M. Andersen[1]

The combination of electronic correlations and Fermi surfaces with multiple nesting vectors can lead to the appearance of complex multi-**Q** magnetic ground states, hosting unusual states such as chiral density waves and quantum Hall insulators. Distinguishing single-**Q** and multi-**Q** magnetic phases is however a notoriously difficult experimental problem. Here we propose theoretically that the local density of states (LDOS) near a magnetic impurity, whose orientation may be controlled by an external magnetic field, can be used to map out the detailed magnetic configuration of an itinerant system and distinguish unambiguously between single-**Q** and multi-**Q** phases. We demonstrate this concept by computing and contrasting the LDOS near a magnetic impurity embedded in three different magnetic ground states relevant to iron-based superconductors—one single-**Q** and two double-**Q** phases. Our results open a promising avenue to investigate the complex magnetic configurations in itinerant systems via standard scanning tunnelling spectroscopy, without requiring spin-resolved capability.

[1] Niels Bohr Institute, University of Copenhagen, Universitetsparken 5, DK-2100 Copenhagen, Denmark. [2] Institut für Theoretische Physik III, Ruhr-Universität Bochum, 44801 Bochum, Germany. [3] Institute of Physics, Kazan Federal University, 420008 Kazan, Russian Federation. [4] School of Physics and Astronomy, University of Minnesota, Minneapolis, Minnesota 55455, USA. Correspondence and requests for materials should be addressed to I.E. (email: Ilya.Eremin@ruhr-uni-bochum.de) or to B.M.A. (email: bma@nbi.ku.dk).

Despite its predominance in localized spin systems[1,2], magnetic frustration is also found in several itinerant systems. While in most cases, magnetic frustration can arise due to the geometry of the lattice or competing exchange interactions, in purely itinerant systems it can be manifested, as a degeneracy among different nesting-driven magnetic instabilities with symmetry-related ordering vectors $\mathbf{Q}$ (ref. 3). Depending on the symmetry of the lattice and on the topology of the Fermi surface, different sets of $\mathbf{Q}$ vectors are possible[4]. In a square lattice, a compensated metal with small hole-like and electron-like Fermi pockets, as shown in Fig. 1a, has magnetic instabilities at the two nesting vectors $\mathbf{Q}_1 = (\pi, 0)$ and $\mathbf{Q}_2 = (0, \pi)$, which are related by a 90° rotation. The resulting double-$\mathbf{Q}$ phases have been shown to also display charge and vector-chirality orders[5]. Such a toy model has been widely employed to study the magnetic properties of iron-based superconductors[6,7] and, more recently, of topological Kondo insulators[8]. In the triangular and honeycomb lattices with a hexagonal Fermi surface, as shown in Fig. 1b, three nesting vectors related by 60° rotations are present, $\mathbf{Q}_1 = (0, \frac{2\pi}{\sqrt{3}})$, $\mathbf{Q}_2 = (\frac{\pi}{3}, -\frac{\pi}{\sqrt{3}})$ and $\mathbf{Q}_3 = (-\frac{\pi}{3}, -\frac{\pi}{\sqrt{3}})$. Interestingly, the possible triple-$\mathbf{Q}$ states display semi-metallic and quantum Hall insulator behaviours. Such a model has been employed to study doped cobaltates and graphene doped to the van Hove singularity point of its band structure[9–12]. Magnetic instabilities of single- versus triple-$\mathbf{Q}$ phases of hexagonal Fermi surface were also studied recently in the context of topological insulators forming magnetic skyrmionic ground-state configurations[13]. In all cases, the presence of repulsive electronic interactions is essential to stabilize the magnetic instabilities over other density wave or superconducting instabilities.

In spite of the interesting properties of multi-$\mathbf{Q}$ phases, unambiguously establishing their existence in a material is a notoriously difficult experimental problem. For instance, neutron scattering, which is the prime tool to probe magnetic configurations, is sensitive not only to the intensity of the order parameters, but also to their corresponding volume fraction. As such, the neutron scattering signatures of a multi-$\mathbf{Q}$ phase can be nearly indistinguishable from the signatures of multiple domains of different single-$\mathbf{Q}$ phases. This general issue has been previously highlighted in the literature in the context of a variety of different materials[14–17]. In some cases, the fact that single-$\mathbf{Q}$ or multi-$\mathbf{Q}$ states break additional discrete symmetries of the lattice may facilitate their experimental distinction. However, in many cases, this distortion may be too small to be resolved experimentally. Several other bulk probes, such as angle-resolved photo-emission spectroscopy, suffer from similar issues.

This analysis begs the question of whether local probes may be more appropriate to distinguish single-$\mathbf{Q}$ versus multi-$\mathbf{Q}$ phases. Indeed, Mössbauer experiments have been recently employed to distinguish single-$\mathbf{Q}$ versus double-$\mathbf{Q}$ phases in iron-based superconductors[18]. Several theoretical calculations[6,19–24] and experimental studies[18,25–34] have revealed that these materials support not only a single-$\mathbf{Q}$ magnetic stripe (MS) phase, with ordering vectors $\mathbf{Q}_1 = (\pi, 0)$ or $\mathbf{Q}_2 = (0, \pi)$, but also two possible types of double-$\mathbf{Q}$ phases: a collinear double-$\mathbf{Q}$ phase with non-uniform Fe-magnetization, called charge-spin density wave (CSDW), and a coplanar double-$\mathbf{Q}$ phase called spin-vortex crystal (SVC)[5]. In the former, the staggered magnetization vectors corresponding to the two $\mathbf{Q}$ vectors are parallel to each other, whereas in the latter they are perpendicular.

In this paper, we propose theoretically that the magnetic field dependence of the local density of states (LDOS) obtained, for example, by performing scanning tunnelling spectroscopy (STS) measurements near a magnetic impurity, allows one to distinguish single-$\mathbf{Q}$ from multi-$\mathbf{Q}$ phases. In a nutshell, the local changes of the spin density amplitude induced around the impurity moment are determined by the specific magnetic ground state of the host. They cause distinct LDOS signatures, since the square of the local spin amplitude couples to the local charge density. We explain in detail this general framework and propose experimental realization for how to manipulate magnetic impurity moments interacting with the surrounding conduction electrons to pinpoint the nature of the magnetic ground state of itinerant systems. We illustrate the proposed set-up in Fig. 2 and focus for concreteness on different proposed magnetic ground states relevant for iron-based superconductors. In the proposed STS experiment shown in Fig. 2, a small fraction of magnetic

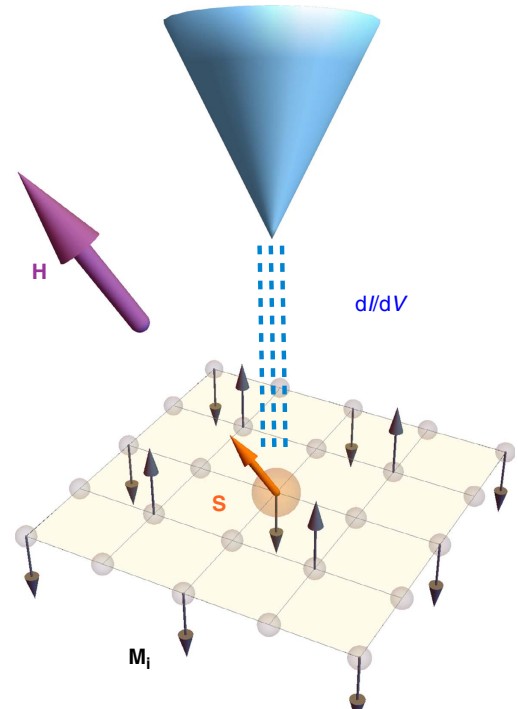

**Figure 2 | Illustration of proposed tunnelling experiment.** The total tunnelling conductance measured by an STS tip (blue tip) at the site of a magnetic impurity moment **S** (orange arrow) embedded in a metallic host of itinerant electrons can be utilized to reveal the detailed magnetic structure of the host magnetism $\mathbf{M_i}$ (black arrows). The external magnetic field **H** is directed along the purple arrow and determines the direction of the impurity moment.

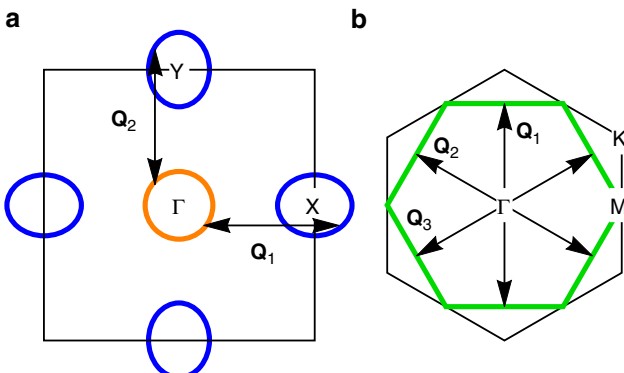

**Figure 1 | Schematic Fermi surfaces.** Illustration of electronic Fermi surfaces with characteristic nesting vectors with tetragonal (**a**) and triangular (**b**) underlying lattice structures.

impurities, such as Mn substitutes for Fe ions in a material that displays one of the possible magnetic configurations discussed above. On the one hand, the orientation of the itinerant ordered magnetic moments of the host system is fixed by the residual spin–orbit and magneto-elastic couplings. On the other hand, the coupling of the magnetic impurity to the host electronic system is determined by the Kondo-like interaction between the impurity and the conduction electrons $J_K$ (refs 35,36), which in the case of Mn in $BaFe_2As_2$ was found to be small by recent electron spin resonance studies[37]. This property, allied to the insensitivity of the itinerant magnetism to external magnetic fields[38,39], implies that magnetic fields are able to 'unlock' the magnetic moment of the impurity from the magnetic order in the lattice, thus allowing for a change of its orientation with respect to the rigid magnetic structure of the host system. In the simplest model, in which the impurity moment orients itself parallel to the applied field, we calculate the impurity-modified LDOS structures as a function of external field direction for all three distinct magnetic ground states. We find qualitative differences in the obtained LDOS spectra and demonstrate that STS offers a promising route to unambiguously distinguish single-$\mathbf{Q}$ from multi-$\mathbf{Q}$ magnetic phases.

## Results

**Single-moment in an itinerant magnet.** We employ a Hamiltonian relevant to iron pnictides, which consists of a five-orbital tight-binding term[40]

$$\mathcal{H}_0 = \sum_{\mathbf{ij},\mu\nu,\sigma} t_{\mathbf{ij}}^{\mu\nu} c_{\mathbf{i}\mu\sigma}^{\dagger} c_{\mathbf{j}\nu\sigma} - \mu_0 \sum_{\mathbf{i}\mu\sigma} n_{\mathbf{i}\mu\sigma}. \qquad (1)$$

Interactions are included through the multi-orbital on-site Hubbard model

$$\begin{aligned} \mathcal{H}_{int} &= U \sum_{\mathbf{i},\mu} n_{\mathbf{i}\mu\uparrow} n_{\mathbf{i}\mu\downarrow} + \left(U' - \frac{J}{2}\right) \sum_{\mathbf{i},\mu<\nu,\sigma\sigma'} n_{\mathbf{i}\mu\sigma} n_{\mathbf{i}\nu\sigma'} \\ &\quad - 2J \sum_{\mathbf{i},\mu<\nu} \mathbf{S}_{\mathbf{i}\mu} \cdot \mathbf{S}_{\mathbf{i}\nu} + J' \sum_{\mathbf{i},\mu<\nu,\sigma} c_{\mathbf{i}\mu\sigma}^{\dagger} c_{\mathbf{i}\mu\bar{\sigma}}^{\dagger} c_{\mathbf{i}\nu\bar{\sigma}} c_{\mathbf{i}\nu\sigma}, \end{aligned} \qquad (2)$$

where $\mu$, $\nu$ are orbital indices, $\mathbf{i}$ denotes lattice sites and $\sigma$ is the spin. The interaction includes intraorbital (interorbital) repulsion $U$ ($U'$), the Hund's coupling $J$ and the pair-hopping term $J'$. Following previous studies[23], we assume spin and orbital rotation invariance, implying $U' = U - 2J$ and $J' = J$, and fix $J = U/4$. As shown previously in ref. 23, the Hamiltonian $\mathcal{H} = \mathcal{H}_0 + \mathcal{H}_{int}$ supports all three magnetic ground states depending on interaction parameters and electron filling. The resulting reconstructed Fermi surfaces in the magnetic phases are remarkably similar, and do not constitute a good probe of the preferred ordered phase (Supplementary Note 1; Supplementary Fig. 1).

The magnetic impurity moment is included by the following term in the Hamiltonian

$$\mathcal{H}_S = \frac{J_K}{2} \sum_{\mu} \mathbf{S}_{\mathbf{i}^*} \cdot \left(c_{\mathbf{i}^*\mu\sigma}^{\dagger} \boldsymbol{\sigma}_{\sigma\sigma'} c_{\mathbf{i}^*\mu\sigma'}\right) - g\mu_B \mathbf{S}_{\mathbf{i}^*} \cdot \mathbf{H}. \qquad (3)$$

The first term corresponds to the Kondo-like exchange coupling between the impurity moment $\mathbf{S}_{\mathbf{i}^*}$, located at site $\mathbf{i}^*$, and the spin of the itinerant electrons, which we denote hereafter by $\mathbf{M}_{\mathbf{i}}^0 \equiv \sum_{\mu} c_{\mathbf{i}\mu\sigma}^{\dagger} \boldsymbol{\sigma}_{\sigma\sigma'} c_{\mathbf{i}\mu\sigma'}$. The second term corresponds to the Zeeman coupling between the impurity moment and the external magnetic field $\mathbf{H}$. In principle, one would also need to include the Zeeman coupling between the itinerant moments and $\mathbf{H}$. However, as shown experimentally, for example, in refs 38,39, the itinerant magnetization is insensitive to magnetic fields of the order of 10 T, and therefore this term can be safely neglected. The main question is whether the external field $\mathbf{H}$ is capable of rotating the impurity moment, which is also coupled to the

itinerant electrons via $J_K$. To answer this question, we rely on electron spin resonance experiments that measured $J_K$ for Mn-doped iron pnictides[37]. The reported estimated value $J_K \approx 1$ meV is very small, suggesting that magnetic fields of the order of 10 T (achievable in STS set-ups) can unlock the impurity moment from the itinerant magnetic configuration. Importantly, the fact that the Mn impurities act as local magnetic moments and do not add charge carriers into the system is supported by the nuclear magnetic resonance data in refs 27,36.

Therefore, we proceed by fixing the direction of $\mathbf{S}_{\mathbf{i}^*}$ to be parallel to $\mathbf{H}$, and introduce the polar angle $\theta$ between the magnetization of the itinerant electrons in the impurity-free system $\mathbf{M}_{\mathbf{i}^*}^0$ and the impurity moment $\mathbf{S}_{\mathbf{i}^*}$ at that site, $\cos\theta \propto \mathbf{S}_{\mathbf{i}^*} \cdot \mathbf{M}_{\mathbf{i}^*}^0$, as illustrated in Fig. 3a. We use a magnetic impurity moment with $J_K S = 0.1$ eV, a value that is not of qualitative importance, since our goal is to focus on the symmetry changes of the LDOS as the impurity moment rotates, that is, as a function of the orientation $\theta$. The optimal magnetic impurity is one that couples strongly to the itinerant electrons leading to a large enough LDOS contrast to be measured, but couples sufficiently weakly that it still 'unlocks' and rotates with the external magnetic field. Thus, we solve equations (1–3) and calculate the LDOS $N(\mathbf{i}, \omega, \theta)$ at each lattice site according to

$$N(\mathbf{i}, \omega, \theta) = -\frac{1}{\pi} \mathrm{Im} \sum_{n\mu\sigma} \frac{u_{\mu\sigma}^n(\mathbf{i}, \theta) u_{\mu\sigma}^n(\mathbf{i}, \theta)}{\omega - E_n(\theta) + i\eta}. \qquad (4)$$

Here $u_{\mu\sigma}^n(\mathbf{i}, \theta)$ are the matrix elements of the unitary transformation from orbital $\mu$ to eigenstate $n$. Below, for concreteness, we focus on the zero energy case $\omega = 0$, and write simply $N(\mathbf{i}, \theta) \equiv N(\mathbf{i}, \omega = 0, \theta)$, but this particular value of $\omega$ is not important for our conclusions. We refer the reader to Supplementary Note 2 for conventional and spin-polarized spectral studies of non-magnetic disorder.

**Impurity-modified LDOS. The CSDW state.** We start by discussing the results in the CSDW state. As illustrated in Fig. 3a,b, in this collinear non-uniform double-$\mathbf{Q}$ magnetic phase the even sites of the square lattice are non-magnetic, whereas the odd sites display a Neel-like magnetic configuration with spins parallel to the $z$ direction. As a result, there are four inequivalent sites to place an impurity, two magnetic ($\mathbf{M}_{\mathbf{i}^*}^0 \neq 0$) and two non-magnetic ($\mathbf{M}_{\mathbf{i}^*}^0 = 0$). Let us start discussing the modified magnetization around the impurity, which will be useful to understand the resulting spectral signatures. The black arrows display the magnetization of the conduction electrons of the impurity-free system $\mathbf{M}_{\mathbf{i}}^0$, and the thick coloured arrow indicates the impurity moment $\mathbf{S}_{\mathbf{i}^*}$. The induced spin density on the nearest neighbour (nn) sites $\mathbf{M}_{\mathbf{nn}}^{ind}$ is illustrated by the four arrows of the same colour. The sum of the induced and impurity-free magnetizations yields the new total magnetization in the presence of the impurity, $\mathbf{M}_{\mathbf{i}} = \mathbf{M}_{\mathbf{i}}^0 + \mathbf{M}_{\mathbf{i}}^{ind}$. In our calculation, the induced spin density actually involves a larger number of sites surrounding the impurity moment, but for the symmetry arguments used below, it is sufficient to focus on the nn sites.

We sketch in Fig. 3e, the change in magnetic moment (projected along the $\hat{\mathbf{l}} = \hat{\mathbf{z}}$ axis) at the four nn sites induced by an impurity oriented along $\theta = 3\pi/4$, $(\Delta\mathbf{M}_{\mathbf{nn}})^2 \equiv (\mathbf{M}_{\mathbf{nn}} \cdot \hat{\mathbf{l}})^2 - (\mathbf{M}_{\mathbf{nn}}^0 \cdot \hat{\mathbf{l}})^2$. Clearly, the magnetization amplitude increases equally at all four nn sites, that is, $|\mathbf{M}_{\mathbf{r}_1} \cdot \hat{\mathbf{l}}| = |\mathbf{M}_{\mathbf{r}_2} \cdot \hat{\mathbf{l}}|$, where $\mathbf{r}_1 = \mathbf{i}^* + \hat{\mathbf{x}}$ and $\mathbf{r}_2 = \mathbf{i}^* + \hat{\mathbf{y}}$ denote the two types of nn sites. If the same impurity moment is placed at a non-magnetic site, however, as illustrated in Fig. 3b, the total projected moments on $\mathbf{r}_1$ and $\mathbf{r}_2$ become unequal. In particular, while the projected moment is reduced at the two sites along the $x$ axis, it is enhanced at the sites along the $y$ axis. This antagonistic change is sketched

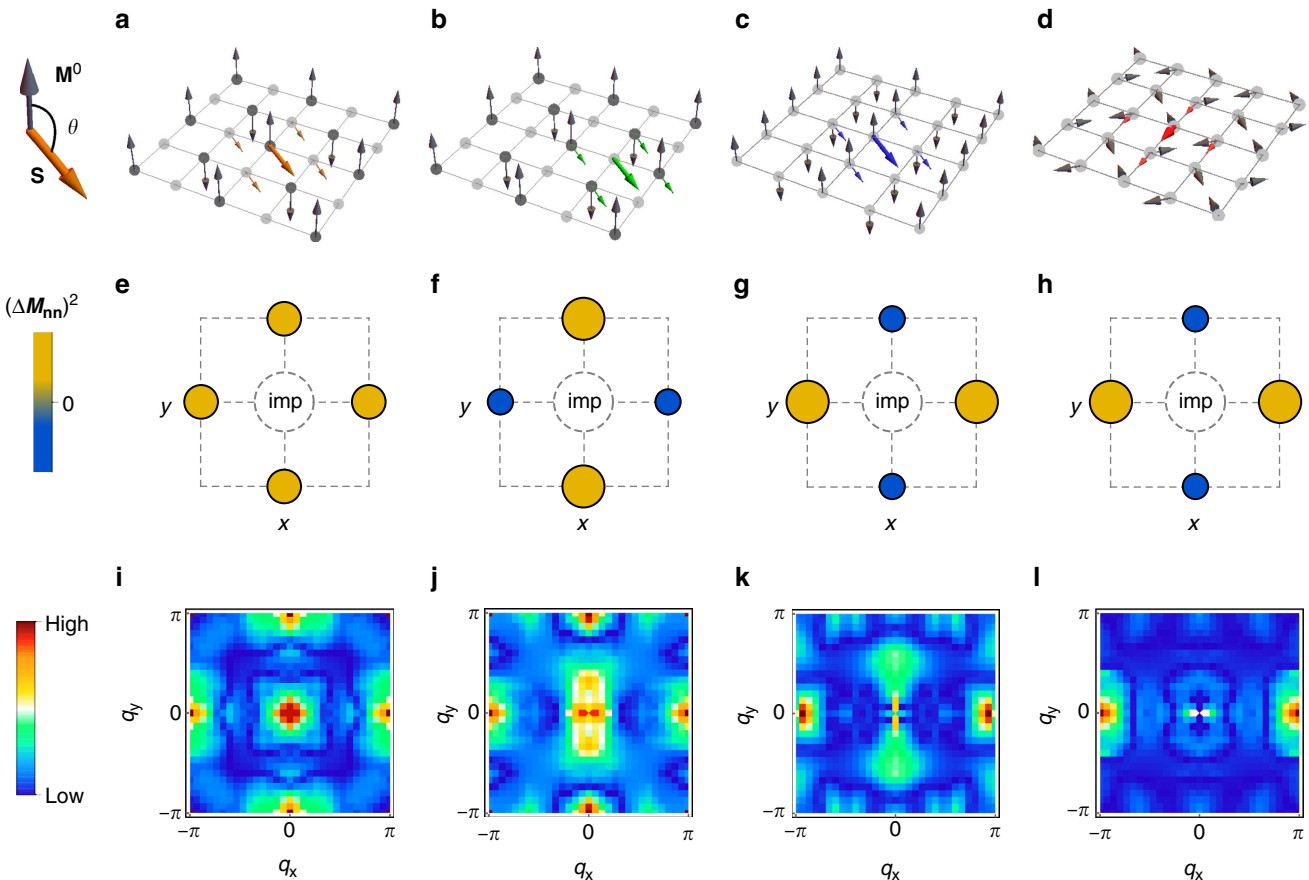

**Figure 3 | Connection between impurity-modified LDOS and the magnetic ground state.** Magnetic impurity (coloured thick arrows) with $\theta = 3\pi/4$ orientation in the CSDW state (either in a magnetic site (**a**) or in a non-magnetic site (**b**)), in the MS state (**c**), and in the SVC state (**d**). The coloured (black) arrows represent the induced (impurity-free) magnetization $\mathbf{M}_{nn}^{ind}$ at the nn sites $\left(\mathbf{M}_i^0\right)$. (**e–h**) Sketches of the total moment amplitude deviation $(\Delta M)^2$ compared with the impurity-free case at the four nn sites $\left((\Delta M_{nn})^2 \equiv \left[(\mathbf{M}_{nn} \cdot \hat{\mathbf{I}})^2 - (\mathbf{M}_{nn}^0 \cdot \hat{\mathbf{I}})^2\right]\hat{\mathbf{I}}\right)$ in the previous four cases for (**e–g**) $\hat{\mathbf{I}} = \hat{\mathbf{z}}$ and (**h**) $\hat{\mathbf{I}} = \hat{\mathbf{y}}$ projection. Yellow (blue) denotes increased (decreased) amplitude. (**i–l**) The Fourier-transformed LDOS $N(\mathbf{q}, \theta = 3\pi/4)$ of the (**a–d**) impurity configurations, respectively.

in Fig. 3f. Consequently, the initial tetragonal symmetry of the magnetization is locally broken by the impurity, with $|\mathbf{M}_{r_1} \cdot \hat{\mathbf{z}}| \neq |\mathbf{M}_{r_2} \cdot \hat{\mathbf{z}}|$. For a general angle $\theta$ between the impurity moment and the itinerant magnetization, this symmetry breaking is given by the following expression,

$$
\begin{aligned}
|\mathbf{M}_{r_1}|^2 - |\mathbf{M}_{r_2}|^2 =\ & \left(\left|\mathbf{M}_{r_1}^{ind}\right|^2 - \left|\mathbf{M}_{r_2}^{ind}\right|^2\right) \\
& - 2\left(\mathbf{M}_{nn}^0 \cdot \hat{\mathbf{z}}\right)\left(\left|\mathbf{M}_{r_1}^{ind}\right| + \left|\mathbf{M}_{r_2}^{ind}\right|\right)\cos\theta.
\end{aligned}
$$
(5)

To make the argument more transparent, we assumed a parallel orientation of $\mathbf{M}_{r_i}^{ind}$ with respect to the impurity moment, which is strictly correct for an arbitrary $\theta$ only in the paramagnetic case, but it remains a reasonable approximation in our case. The full self-consistent result beyond this assumption is presented in Fig. 4.

To understand the distinct changes of the surrounding magnetization, we note that impurities at the magnetic sites of the CSDW phase (Fig. 3a) are subject to two constrains. First, the impurity-free SDW has nodes at the nn sites $\left(\mathbf{M}_{nn}^0 \cdot \hat{\mathbf{z}} = 0\right)$, and second, the symmetry of this site requires the induced moments to be the same, $\mathbf{M}_{r_1}^{ind} = \mathbf{M}_{r_2}^{ind}$. Hence, $|\mathbf{M}_{r_1}|^2 = |\mathbf{M}_{r_2}|^2$ and tetragonal symmetry is preserved for any orientation of the impurity moment. By contrast, for impurity moments at non-magnetic sites, the difference between the total nn amplitudes has

contributions from both terms in equation (5). Therefore, the symmetry-breaking expression is generally non-zero, and exhibits a cosine-like directional dependence.

Having established the basic effect of an impurity moment on the surrounding itinerant magnetic structure, we now study its consequences on the local spectral features. The total LDOS measures the spectral composition of the charge density, which is coupled by symmetry to the magnetization density squared. Therefore, any change in the amplitude of the spin density will have an impact on the LDOS. We use this simple argument for the sake of simplicity, and stress that strictly speaking it is only valid in the paramagnetic phase, since a classical magnetic impurity directly generates also a charge modulation in the SDW phase (that is, the charge modulations are not just slaved to the induced additional spin polarization in the SDW phase). For instance, a $C_4$ ($C_2$) symmetric inhomogeneous magnetic structure will generally present $C_4$ ($C_2$) symmetric LDOS signatures (here $C_4$ and $C_2$ denote tetragonal and orthorhombic symmetries). Figure 3i shows the Fourier-transformed LDOS of the case displayed in Fig. 3a, $N(\mathbf{q}, 3\pi/4)$. Clearly, the $C_4$ symmetry of the pristine CSDW state is preserved around the impurity, in agreement with the magnetic moment structure sketched in Fig. 3e and the result $|\mathbf{M}_{r_1}|^2 = |\mathbf{M}_{r_2}|^2$ given by equation (5). The spectral symmetry is in fact tetragonal for all possible orientations of the impurity in the CSDW phase, as long as the impurity moment is placed at a magnetic site. If the same impurity

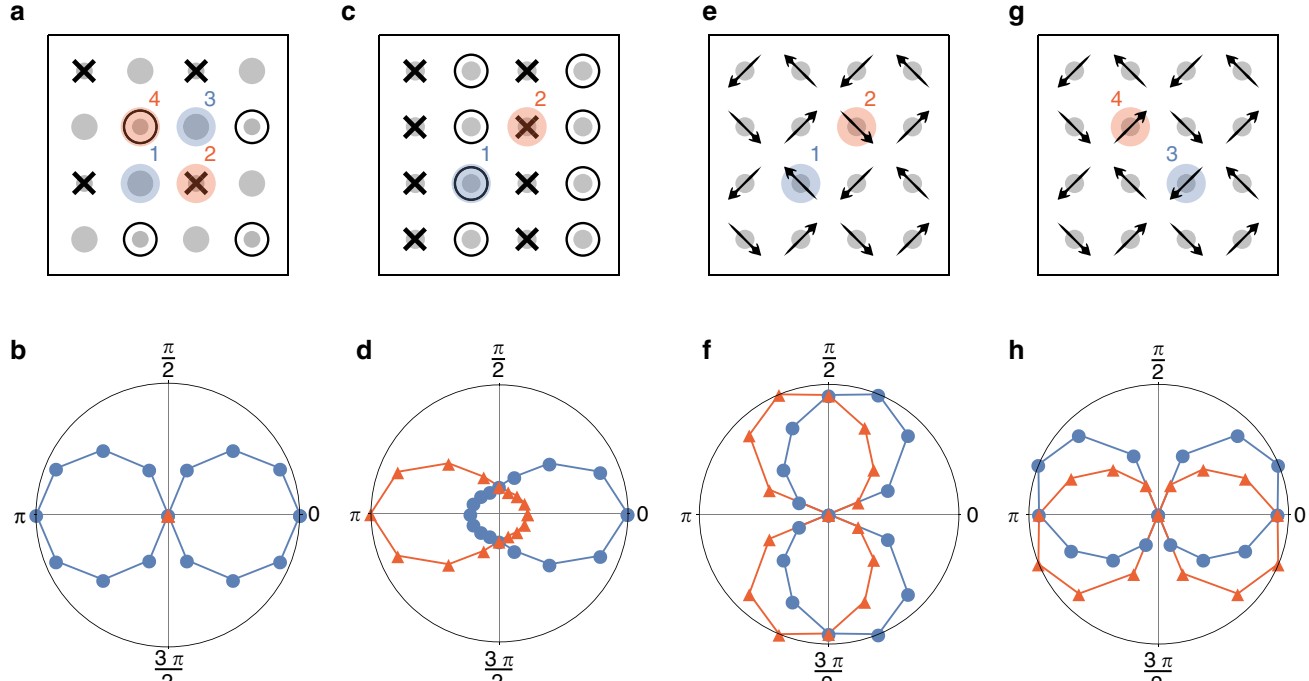

**Figure 4 | Angular dependence of $C_4$ symmetry breaking in the LDOS.** Spectral symmetry-breaking parameter $\delta_{C_2}(\theta)$ (equation (6)) as a function of the orientation of the magnetic impurity $\theta$ in the inequivalent sites of the magnetic states. $\theta$ represents the polar angle in the (**a**,**b**) CSDW and (**c**,**d**) MS states, and (**e**–**h**) the azimuthal angle in the SVC state. $\theta = 3\pi/4$ examples are explicitly shown in Fig. 3. (**b**,**d**,**f**,**h**) are polar plots of the average $\delta_{C_2}(\theta)$ as radius in a given direction $\theta$ in the $xz$ ($xy$) plane for all relevant inequivalent sites of the CSDW and MS (CSV) states.

moment is placed at a non-magnetic site, however, as illustrated in Fig. 3b, $N(\mathbf{q}, 3\pi/4)$ becomes $C_2$ symmetric as seen in Fig. 3j. This is simply a consequence of the $C_2$ symmetric spin structure induced by the impurity discussed earlier and sketched in Fig. 3f.

To quantify the evolution of the spectral symmetry of the LDOS as the impurity rotates, and compare it with equation (5), we introduce the anisotropy parameter

$$\delta_{C_2}(\theta) = \sum_{\mathbf{i}} \frac{|N(\mathbf{i}, \theta) - N(R\{\mathbf{i}\}, \theta)|}{N(\mathbf{i}, \theta) + N(R\{\mathbf{i}\}, \theta)}, \qquad (6)$$

which measures the breaking of the $C_4$ symmetry for a given $\theta$. Here, $R$ denotes a $\pi/2$ rotation operation. Figure 4b shows the evolution of $\delta_{C_2}$ as a function of $\theta$ from the calculated LDOS at all four inequivalent sites in the CSDW state, specified in Fig. 4a. As seen, tetragonal symmetry is indeed preserved for all $\theta$ at the magnetic sites (2 and 4), but broken in a cosine-like manner at the non-magnetic sites (1 and 3), in agreement with the above discussion.

The MS state. The single-$\mathbf{Q}$ MS phase corresponds to the case, in which only one of the two possible $\mathbf{Q}$ vectors is selected. The corresponding magnetic configuration, shown in Fig. 3c, breaks the $C_4$ symmetry down to $C_2$, in contrast to the double-$\mathbf{Q}$ magnetic configurations, which preserve tetragonal symmetry. We now consider the effects of an impurity moment with $\theta = 3\pi/4$ orientation with respect to the SDW magnetization. The inequivalent change in the spin projection along the $z$ axis of the four nn sites is illustrated in Fig. 3g, which accounts for the symmetry breaking in the corresponding $N(\mathbf{q}, 3\pi/4)$ shown in Fig. 3k. The evolution of $\delta_{C_2}(\theta)$ for the two inequivalent sites in this state can be seen in Fig. 4d. Clearly, both sites (specified in Fig. 4c) give rise to distinct angular evolutions that break $C_2$ symmetry. This is in contrast to what was found in the CSDW state, where half of the sites exhibit a $C_2$ symmetric cosine-like

evolution of $\delta_{C_2}(\theta)$, and the other half preserve $C_4$ symmetry for all impurity orientations, $\delta_{C_2}(\theta) = 0$.

The SVC state. Finally, we discuss the case of a magnetic impurity in the coplanar SVC phase. In this double-$\mathbf{Q}$ magnetic state, the even sites of the square lattice display a Neel-like order that is perpendicular to the Neel-like order displayed by the odd sites. As a result, there are four inequivalent sites. One of them is shown in Fig. 3d, with the calculated LDOS displayed in Fig. 3l. Again, the $C_2$ symmetric spectral features can be connected to the different moment amplitudes at the nn sites, as illustrated in Fig. 3h for projections along the $y$ axis. The angular evolution of $\delta_{C_2}(\theta)$ in the four inequivalent sites (in the $xy$ plane for this state) is shown in Fig. 4f,h. Contrary to what was found in the CSDW phase, all inequivalent sites show a $\delta_{C_2}(\theta)$ spectral angular dependence with broken $C_2$ symmetry. Moreover, the angular dependences appear identical, but shifted by $n\pi/2$ for all four inequivalent sites ($n = 1, 2, 3$), which is a consequence of the uniform moment amplitude $|\mathbf{M}_{\mathbf{i}}^0| = M^0$ in the SVC state.

## Discussion
The results presented in the previous section open different routes to distinguish between single-$\mathbf{Q}$ and multi-$\mathbf{Q}$ magnetic ground states via STS experiments. The most direct way would be to extract the LDOS anisotropy parameter $\delta_{C_2}$ as function of the angle $\theta$ between the applied magnetic field and the magnetization. Note that a recent STS experiment on the iron pnictide NaFeAs extracted precisely this anisotropy parameter (for zero magnetic field)[41]. As shown in Fig. 4, the function $\delta_{C_2}(\theta)$ behaves qualitatively different for each of the three magnetic ground states.

First, in the CSDW state, experiments would only observe a single anisotropy parameter curve $\delta_{C_2}(\theta)$, which displays a cosine-like $C_2$ symmetric shape (Fig. 4b). This curve corresponds

to a magnetic impurity placed in a non-magnetic site, since impurities located at magnetic sites exhibit fully $C_4$-symmetric LDOS.

Second, in the MS state, experiments would observe two anisotropy parameter curves $\delta_{C_2}(\theta)$ (Fig. 4d). These two curves, related to the two inequivalent site positions in the magnetic ground state, are simply related by a $\theta = \pi$ shift. In addition, tetragonal symmetry is broken for *all* impurity orientations, that is, $\delta_{C_2}(\theta) \neq 0$ for all $\theta$ values.

Third, in the SVC state, experiments would observe four anisotropy parameter curves $\delta_{C_2}(\theta)$ (Fig. 4f,h). They also correspond to the four inequivalent site positions of the SVC state, and are related to each other by successive shifts of $\theta = \pi/2$.

The proposed experiment advocated in this paper relies on the fact that an impurity moment can be controlled by an external applied field which, however, does not significantly affect the itinerant magnetic order. In addition, it would be desirable if this impurity did not introduce additional charge carriers in the system. In the case of the iron pnictides, Mn impurities are natural candidates, since they form local moments weakly coupled to the itinerant system[27,36,37], which is itself robust against moderate magnetic fields[38]. It will be interesting to extend our calculations to multi-**Q** magnetic phases in triangular lattices, where exotic triple-**Q** phases can appear. Our work provides a promising avenue for future tunnelling spectroscopy to directly distinguish between nearly degenerate but symmetry-distinct magnetic ground states of itinerant magnetic systems.

## Methods

**Computational details**. A mean field decoupling in the spin and charge channels leads to the following total Hamiltonian

$$\mathcal{H}^{MF} = \sum_{\mathbf{ij}\mu\nu} \begin{pmatrix} \hat{c}^\dagger_{\mathbf{i}\mu\uparrow} & \hat{c}^\dagger_{\mathbf{i}\mu\downarrow} \end{pmatrix} \begin{pmatrix} \varphi^{\mu\nu}_{\mathbf{ij}\uparrow} & \omega^{\mu\nu}_{\mathbf{ii}\uparrow} \\ \omega^{\mu\nu}_{\mathbf{ii}\downarrow} & \varphi^{\mu\nu}_{\mathbf{ij}\downarrow} \end{pmatrix} \begin{pmatrix} \hat{c}_{\mathbf{j}\nu\uparrow} \\ \hat{c}_{\mathbf{j}\nu\downarrow} \end{pmatrix}, \tag{7}$$

where $c^\dagger_{\mathbf{i}\mu\sigma}$ creates an electron at site $\mathbf{i}$ with spin $\sigma$ in orbital state $\mu$. $\varphi^{\mu\nu}_{\mathbf{ij}\sigma}$ and $\omega^{\mu\nu}_{\mathbf{ii}\sigma}$ are given by

$$\begin{aligned} \varphi^{\mu\nu}_{\mathbf{ij}\sigma} &= t^{\mu\nu}_{\mathbf{ij}} + \delta_{\mu\nu}\Big[-\mu_0 + \big(\Omega^0_\mu + \Omega^z_\mu\big)\delta_{\mathbf{ii}^*}\delta_{\mu\nu} + U\langle\hat{n}_{\mathbf{i}\mu\overline{\sigma}}\rangle \\ &\quad + U'\langle\hat{n}_{\mathbf{i}\nu\overline{\sigma}}\rangle + (U'-J)\langle\hat{n}_{\mathbf{i}\nu\sigma}\rangle\Big] \\ &\quad - \overline{\delta}_{\mu\nu}\Big[(U'-J)\langle\hat{c}^\dagger_{\mathbf{i}\nu\sigma}\hat{c}_{\mathbf{i}\mu\sigma}\rangle - J\langle\hat{c}^\dagger_{\mathbf{i}\nu\overline{\sigma}}\hat{c}_{\mathbf{i}\mu\overline{\sigma}}\rangle\Big] \\ &\quad - J'\langle\hat{c}^\dagger_{\mathbf{i}\mu\overline{\sigma}}\hat{c}_{\mathbf{i}\nu\overline{\sigma}}\rangle\Big], \end{aligned} \tag{8}$$

$$\begin{aligned} \omega^{\mu\nu}_{\mathbf{ii}\sigma} &= \delta_{\mu\nu}\Big[\Omega^x_\mu\delta_{\mathbf{ii}^*}\delta_{\mu\nu} - U\langle\hat{c}^\dagger_{\mathbf{i}\mu\sigma}\hat{c}_{\mathbf{i}\mu\sigma}\rangle - J\langle\hat{c}^\dagger_{\mathbf{i}\nu\overline{\sigma}}\hat{c}_{\mathbf{i}\nu\sigma}\rangle\Big] \\ &\quad - \overline{\delta}_{\mu\nu}\Big[U'\langle\hat{c}^\dagger_{\mathbf{i}\nu\overline{\sigma}}\hat{c}_{\mathbf{i}\mu\sigma}\rangle + J'\langle\hat{c}^\dagger_{\mathbf{i}\mu\overline{\sigma}}\hat{c}_{\mathbf{i}\nu\sigma}\rangle\Big], \end{aligned} \tag{9}$$

with $\overline{\delta}_{\mu\nu} = 1 - \delta_{\mu\nu}$. The terms $\Omega^z_\mu = \sigma J_K \mathbf{S}_\mu \cdot \hat{\mathbf{z}}$ and $\Omega^x_\mu = J_K \mathbf{S}_\mu \cdot \hat{\mathbf{x}}$ include out-of-plane and in-plane components of a magnetic impurity at site $\mathbf{i}^*$, respectively. A non-magnetic scatterer can be also introduced at the same site by the $\Omega^0_\mu = V_\mu$ term. In the main text, this last term was set to zero, but for a brief non-magnetic impurity study the reader is refered to Supplementary Note 2. We diagonalize equation (7) on $30 \times 30$ lattices by a unitary transformation $\hat{c}_{\mathbf{i}\mu\sigma} = \sum_n u^n_{\mu\sigma}(\mathbf{i})\hat{\gamma}_n$, and the following unrestricted fields are obtained self-consistently:

$$\begin{aligned} \langle\hat{c}^\dagger_{\mathbf{i}\mu\sigma}\hat{c}_{\mathbf{i}\nu\sigma}\rangle &= \sum_n u^{n*}_{\mu\sigma}(\mathbf{i})u^n_{\nu\sigma}(\mathbf{i})f(E_n), \\ \langle\hat{c}^\dagger_{\mathbf{i}\mu\sigma}\hat{c}_{\mathbf{i}\nu\overline{\sigma}}\rangle &= \sum_n u^{n*}_{\mu\sigma}(\mathbf{i})u^n_{\nu\overline{\sigma}}(\mathbf{i})f(E_n), \end{aligned} \tag{10}$$

for all sites $\mathbf{i}$, spin $\sigma$ and orbital combinations $\mu$, $\nu$. Here $E_n$ denote the eigenvalues and $f$ is the Fermi function. The magnetization density $\mathbf{M_i} = \sum_{\mu\sigma\sigma'}\langle\hat{c}^\dagger_{\mathbf{i}\mu\sigma}\boldsymbol{\sigma}_{\sigma\sigma'}\hat{c}_{\mathbf{i}\mu\sigma'}\rangle$ ($\mu_B = 1$, $g = 2$) and charge density $n_{\mathbf{i}} = \sum_{\mu\sigma}\langle\hat{c}^\dagger_{\mathbf{i}\mu\sigma}\hat{c}_{\mathbf{i}\mu\sigma}\rangle$ are obtained from the self-consistent fields in equation (10). For further computational details, see Supplementary Note 1. All relevant data are available from the authors.

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

## Acknowledgements

M.N.G. and B.M.A. acknowledge support from Lundbeckfond fellowship (grant A9318). R.M.F is supported by the Office of Basic Energy Sciences, U.S. Department of Energy, under award DE-SC0012336. I.E. was supported by the joint DFG-ANR Project (ER 463/8-1). I.E. also acknowledges the support from the project of the state assignment of KFU in the sphere of scientific activities.

## Author contributions

All authors participated in the overall construction of the project, the discussion of the results, and in the writing of the paper. M.N.G. performed all analytical and numerical calculations. The whole project was supervised by B.M.A. All authors have read and approved the final version of the manuscript.

## Additional information

**Competing financial interests:** The authors declare no competing financial interests.

**Publisher's note**: 

