## [Peer Review File · Nature Communications]

Reviewers' comments:

Reviewer #1 (Remarks to the Author):

Gastiasoro et al. propose an STM methodology to discern the ground states of itinerant magnets. Having iron-based superconductors in mind, they consider a tetragonal lattice that may host magnetic stripe (MS), charge-spin density-wave (CSDW) and spin-vortex crystal (SVC) phases. The basic idea is that a magnetic impurity, of which magnetic moment can be rotated by external magnetic field, may generate characteristic angular-dependent local-density-of-states (LDOS) patterns that reflect the magnetic structures of the host itinerant magnets.

The topic of the manuscript is interesting because distinguishing different ground states is important, in general. I appreciate the proposed method but have some concerns/comments as listed below.

1. I think that MS, CSDW and SVC phases can be distinguished even without having an impurity. MS breaks C_{4v} symmetry of the lattice while the others maintain it. The CSDW phase has (n, n) superstructure of the charge that is absent in the SVC phase. Thus each phase has its own feature that should manifest in the STM topography and/or LDOS patterns imaged by standard STM/STS technique. Related comment: the authors write in the supplementary material that "The amplitude of this (CSDW) modulation, however, may be too small to be easily detected by tunneling spectroscopy." I suggest the authors to give more quantitative argument on this.

2. It looks to me that the angular-dependent anisotropy parameter (Figs. 4 b,d,f,h) is associated with the local symmetry of the impurity site. If so, even nonmagnetic impurity can be used to distinguish the host phases. The authors actually argue the effects of nonmagnetic impurities in the supplementary material but the impurity site is not specified. As described in the main text, there are inequivalent impurity sites in each phase. I recommend the authors to include the discussion on the effects of nonmagnetic impurities at different sites in the main text.

3. The authors write in the discussion section that, even if field rotation is not possible, the change in the field direction from positive to negative may be useful to distinguish the different host phases. However, the field should be applied along the perpendicular and parallel to the plane for the CSDW and SVC phases, respectively; one still need to rotate the field direction by 90 degrees to distinguish these two phases. Please comment on this.

Reviewer #2 (Remarks to the Author):

Gastiasoro et al focus on a challenging and fundamental problem of condensed matter physics, identifying a Fermi surface (FS) generated, multi-Q, spin density wave (SDW).

When a SDW is generated by FS nesting, there are multiple wavevectors (Q) related by discrete symmetries of the electronic band structure. It has proven extremely difficult to discriminate single-Q multi-domain from multi-Q single-domain SDW states, and this is especially important in compounds where iron-based high temperature superconductivity appears. Gastiasoro et al introduce an ingenious new local probe approach to solve this problem. It is based on use of field dependent scanning tunneling spectroscopy (STS) plus control of the quantization axis of the spin state of an impurity atom at the origin, Mn being the discussed. The Hamiltonian and band structure used is a model of iron-based superconducting compounds and, in that sense, is specialized but highly relevant. The impurity Hamiltonian (Eqn. 3) is familiar.

Now, the crux of this proposal is to experimentally manipulate the impurity spin quantization axis in real-space relative to the spin density modulations of the SDW, by using reorientable magnetic field. For various SDW models the authors establish the resulting perturbation of site-specific magnetization (compared to the background order with no field) nearby; this is quite impressive and convincing in Fig 3a,b,c,d. If one could directly detect the site specific magnetization here, this problem would be solved in general. So to this point Gastiasoro et al introduce a fascinating and excellent new proposal.

The next step is to link perturbation of site-specific magnetization to perturbation of site-specific local density of states LDOS(r). Unfortunately, at least as far I can understand it, the link to experiment breaks down here. Authors state "The total LDOS measures the spectral composition of the charge density, which is coupled by symmetry only to the magnetization density squared. Therefore, any change in amplitude of the spin density will have an impact on the LDOS" etc. While true in principle, for most energies spanning all the bands of a material their amplitude of this perturbation is infinitesimally small. Such effects only become significant in some narrow energy range of coupling J_K and only for specific delocalized states that are coupled strongly to the impurity via Eqn 3.

At a practical level this is key because what is measurable as LDOS $N(r,E)$ is $dI/dV(r,E)$ where dI/dV is the differential tunnel conductance of STM tip to sample at a given site and bias voltage $V=E/e$, and this is unavoidably an energy resolved quantity. But to best of my understanding Eqn 4 and 6 are not energy resolved (or if they are this is not made clear & specific). Instead they appear to span whole band(s) thus being the site specific charge & no such measurement of site-specific net charge has even been reported.

What is measurable is the energy-resolved and site-resolved version of Eqn 6 and Fig 4a,c,e,g. For example, an energy resolved version of Eqn 6, where site-specific C4 breaking perturbations are only due to intra-unit-cell charge redistributions in a charge ordered state, have been reported in e.g. Nature 466, 374 (2010); Science 333, 426 (2011); J. Phys. Soc. Jpn. 81, 011005 (2012); New J. Phys. 14 053017 (2012); Proc. Nat. Acad. Sci. 111, E3026 (2014) and Nature Physics 12, 150 (2016). So, for this elegant and ingenious proposal to have strongest impact on the field, it is the site-specific energy-resolved $N(r,E)$ and the resulting energy-resolved Eqn 6 surrounding the reorientable magnetic impurity that is key. Given a version of Eqn. 6 and Fig 4 which identifies for experimenter how to determine the correct energies (measured from chemical potential) where the beautiful effects shown in Fig 4a,c,e,g and evaluated in an elegant new fashion in Fig 4b,d,f,h have their maximum intensity, Gastiasoro et al could then become profoundly impactful.

With such adjustments and/or clarifications, I would be enthusiastic to recommend this manuscript for publication.

Reviewer #3 (Remarks to the Author):

In this manuscript authors propose a new method in order to distinguish multi-Q magnetic phases in itinerant magnetic systems. The idea is based on the different polarization effects of magnetic impurities on the moments of neighboring sites which depends on the underlying magnetic structure and on the orientation of the impurity moment. Authors have in mind iron-based superconductors with a small number of magnetic impurities, e.g. Mn ions, replacing Fe. Experimentally, the holes from the Mn impurities do not delocalize and only act as local moments which couple to the itinerant

magnetic system via a Kondo-like exchange coupling J_K .

The proposal builds on the assumption that

- (a) the external magnetic field does not change the magnetic structure.
- (b) the local moment can be controlled by an external field, and
- (c) the impurity does not influence on the underlying magnetic structure.

Concerning (a), authors present evidence from the literature (Refs. 38,39) while (b) and (c) require J_K to be 'sufficiently small' which seems to be supported by the ESR experiments of Ref. 37. On the other hand, induced magnetic moments M_{ind} then are also small and even more the LDOS variations which depend on M_{ind}^2 . In this regard I don't agree with the statement on page 3 that the value of the magnetic impurity moment $J_k S$ "is not important". It matters for the expected contrast which in Fig. 3 is only labeled as 'low' and 'high'. Experimentally, the detection of these small variations may be complicated by the fact that the 'natural' cleavage plane of 122 compounds is between the alkaline and As layers whereas Fe terminations are rare (Li et al., arXiv:1602.04937). Concerning (c), authors refer to the literature.

Authors use a 5 orbital tight-binding model for the electronic structure of iron-pnictides. However, this model is in principle derived from Wannier functions which relate the field operator $\Psi(R)$ to site ' r_i ' in the tight-binding model within a certain range $|R-r_i|$ given by the extension of the Wannier orbital (Choubey et al., PRB 90, 134520 (2014)). The analysis in the present manuscript sensitively depends on the LDOS variations on sites close to the impurity which can be significantly smeared or altered by including the structure of Wannier orbitals and therefore spoil the predicted angular dependence of the C_2 anisotropy parameter.

In summary, the work by Gastiasoro is very appealing and presents an interesting approach for detecting multi-Q magnetic phases. However, based on the arguments given above I think that the present stage of the theory is on a very simplified model level and not yet as convincing as required for publication in Nature Communications.

REVIEWERS' COMMENTS:

Reviewer #1 (Remarks to the Author):

The replies to the reviewers' comments reasonably address most of the issues, even though the authors made little change in the main text. I would recommend the publication of this manuscript if the following minor issues would be addressed.

1. I appreciate that the authors are proposing a new framework for solving the single-Q vs. multi-Q problem. Nevertheless, because a particular symmetry is broken in each case, conventional STM/STS technique might also be able to discern different ground states. Therefore the authors should show at least one example where the conventional technique may not work well; otherwise alternative technique might not be necessary. In the reply, the authors present a couple of good examples (cuprates and pnictides) in which CDW order is hard to be detected by conventional STM/STS. I recommend the authors to describe these examples in the main text to make clear the potential of the authors' method.

2. As for the magnetic-field rotation, from experimental point of view, it is quite easy to reverse the field direction (0 and π) but it is challenging to switch the field direction from 0 to $\pi/2$ because two independent magnets should be installed in the system, or the STM should be rotatable. Nevertheless, these techniques are not impossible. If there are two magnets, a magnetic field can be applied along arbitrary direction by controlling the field strengths of the two magnets. (The highest field is limited to a few tesla.) If the STM is rotatable, field direction control is easy, of course. Thus continuous rotation is as easy as "0 and $\pi/2$ " and there is no particular advantage in the latter. I recommend the authors just to write that the field rotation is not easy but possible.

Reviewer #2 (Remarks to the Author):

I thank authors for very clear expositions and improvements to manuscript in answer to my questions . I now find this paper eminently appropriate for immediate publication.

Reviewer #3 (Remarks to the Author):

Authors have answered in detail to all points raised by the referees and have also in some parts modified the manuscript accordingly. My basic assessment of the work is still ambivalent. On the one hand I really appreciate the interesting proposal for distinguishing single- and multi-Q magnetic states which is new and definitely deserves publication in some form. On the other hand, from the results presented in the manuscript it is still not clear to me if the method can be experimentally realized. This realization requires a coupling J_K of the impurity to the itinerant moments which is small enough that magnetic fields can "unlock the impurity moment from the itinerant magnetic configuration" (lines 170,171). However, the induced LDOS variations are proportional to the squared induced magnetization and thus also rather small. In the manuscript, authors report a value of $J_K \approx 0.8$ meV (line 168) but perform calculations with $J_K S=100$ meV (and 10 meV in the reply to referee #2). What would be the contrast for the realistic, much smaller value?

In this context referee #1 raised the interesting point that in principle a nonmagnetic impurity (which couples directly to the density) could also distinguish the different phases. Authors show the resulting LDOS patterns for the three phases and two impurity sites in Fig. 1 of their reply. In fact, this allows to distinguish MS (C2-symmetry) from SVC and CSDW (C4-symmetry) where the latter phases can in addition be distinguished due to the inequivalent sites in the CSDW phase. Now authors argue that the corresponding contrast is small because it depends on the CDW amplitude differences. Is it really smaller than the contrast obtained with the magnetic impurity and a coupling of $J_K \approx 0.8$ meV ?

This lacking of an estimate for the expected contrast in an experimental realization is my only concern.

The principle proposed in the manuscript is definitely new and interesting and may nevertheless challenge colleagues in the field to perform the experiment.

Reviewer #1 (Remarks to the Author):

The referee writes:

“Gastiasoro et al. propose an STM methodology to discern the ground states of itinerant magnets. Having iron-based superconductors in mind, they consider a tetragonal lattice that may host magnetic stripe (MS), charge-spin density-wave (CSDW) and spin-vortex crystal (SVC) phases. The basic idea is that a magnetic impurity, of which magnetic moment can be rotated by external magnetic field, may generate characteristic angular-dependent local-density-of-states (LDOS) patterns that reflect the magnetic structures of the host itinerant magnets.

The topic of the manuscript is interesting because distinguishing different ground states is important, in general. I appreciate the proposed method but have some concerns/comments as listed below.

1. I think that MS, CSDW and SVC phases can be distinguished even without having an impurity. MS breaks C₄ symmetry of the lattice while the others maintain it. The CSDW phase has (π, π) superstructure of the charge that is absent in the SVC phase. Thus each phase has its own feature that should manifest in the STM topography and/or LDOS patterns imaged by standard STM/STS technique. Related comment: the authors write in the supplementary material that “The amplitude of this (CSDW) modulation, however, may be too small to be easily detected by tunneling spectroscopy.” I suggest the authors to give more quantitative argument on this.”

Answer:

We thank the referee for a series of excellent questions. In principle the referee is correct, but there are serious practical challenges that obscure utilizing the homogeneous phases as a probe of the magnetic structure. In fact this is the main motivation for our study. For example, for the MS, it is true that it breaks the tetragonal symmetry of the lattice. However, domains make it very hard to distinguish it from the double-Q phases, particularly if the associated lattice distortion is small (see for instance Phys. Rev. B 90, 174511 (2014)). Moreover, bulk measurements like transport, neutron scattering etc. present an average of any x-y anisotropy in the system arising from different domains, which again hampers their identification of the MS phase.

Regarding use of the CDW order in the CSDW phase to distinguish it from SVC, then we have not emphasized its properties and potential detectability because it is a peculiarity of the CSDW phase of the pnictides; we are proposing a general framework for solving the single-Q vs. multi-Q problem of any itinerant system. We note, however, that experience from the cuprates has proven that clear STM signatures from CDW order can be very hard to detect experimentally. For the particular CDW (π, π) super-structure in pnictides, we note also that (π, π) is a Bragg peak because of the Fe-As hybridization, making it potentially harder to detect the CDW structure since it falls on top of a Bragg peak. For these reasons we believe that the proposal to use a magnetic tunable moment is more robust and unambiguous as a means to nail down the preferred order.

The referee writes:

“2. It looks to me that the angular-dependent anisotropy parameter (Figs. 4 b,d,f,h) is associated with the local symmetry of the impurity site. If so, even nonmagnetic impurity can be used to distinguish the host phases. The authors actually argue the effects of nonmagnetic impurities in the supplementary material but the impurity site is not specified. As described in the main text, there are inequivalent impurity sites in each phase. I recommend the authors to include the discussion on the effects of nonmagnetic impurities at different sites in the main text.”

Answer:

Nonmagnetic impurities couple directly to the charge density. Both MS and SVC phases have homogeneous charge densities in the absence of impurities, i.e. the local symmetry in terms of charge density (and thus DOS) is the same everywhere in the unperturbed systems, and therefore nonmagnetic potentials placed in any site will show identical $N(i, \omega)$, as shown below in Fig. 1(a) for the MS state and Fig. 1(b) for the SVC state. That is, a given potential V will show the same C2 symmetric feature $N(i, \omega)$ at any site i in the MS state, and the same C4 symmetric feature at any site in the SVC state.

In the case of the CSDW state, there are two inequivalent charge density sites arising from the associated (π, π) CDW of the unperturbed system, and therefore this state will show two different C4 symmetric LDOS patterns around potentials, as illustrated in Fig. 1(c). The referee is correct that in principle this fact could be used to distinguish CSDW and SVC phases, but it relies again on the small “slaved” CDW order (see discussion above). The distinction between these two features, shown also in momentum space in Fig. 1(d), depends, therefore, on amplitude differences comparable to the CDW amplitude. We have expanded the SM material discussing the nonmagnetic impurities but prefer not to include a discussion of this in the main text since it is not the focus of our more generally applicable setup for identifying single- Q and multi- Q order.

The main point of the paper is to exploit the coupling of magnetic impurities to the spin degrees of freedom of itinerant electrons of a given magnetic state, which, as explained in the main text, shows overall symmetry breaking features upon manipulation of the single-moment that can be tracked back to the magnetic ground state of the system.

Figure 1. $N(i, \omega=-25\text{meV})$ for a $V=100\text{ meV}$ nonmagnetic impurity in two nearest neighbor sites denoted (I) and (II) as inset. (a) MS state with $M^0_i \cdot \hat{z} > 0$ at site (I) and $M^0_i \cdot \hat{z} < 0$ at site (II). (b) SVC state at sites with orthogonal M^0_i . (c) CSDW state at sites with $M^0_i = 0$ for site (I) and $M^0_i = 0$ for site (II). (d) The Fourier transform $N(q, \omega=-25\text{meV})$ of cases shown in (c). The (π, π) peaks have been removed from the color scale for clarity.

The referee writes:

“3. The authors write in the discussion section that, even if field rotation is not possible, the change in the field direction from positive to negative may be useful to distinguish the different host phases. However, the field should be applied along the perpendicular and parallel to the plane for the CSDW and SVC phases, respectively; one still need to rotate the field direction by 90 degrees to distinguish these two phases. Please comment on this.”

Answer:

We meant a continuous rotation of the magnetic field from $\theta=0$ to π in the relevant plane for each magnetic state (where the itinerant spins M_i are oriented). I.e. “if a continuous field rotation is not possible..”. Experimentally it might be challenging to access multiple θ orientations. We just wanted to point out that two or three orientations can be enough to deduce the true ground state, as illustrated in Fig. 2 below. There it is shown explicitly that e.g. $\theta=\pi/2$ and $\theta=0$ would be enough to unambiguously identify the ground state magnetic order since the three states exhibit different values of δ_{C_2} at these angles. For example, CSDW and MS is distinguishable by the finite value at $\theta=\pi/2$ in the MS state, and CSDW and SVC are distinguishable by the finite value at half the sites at $\theta=\pi/2$ in the SVC phase. We have reformulated the relevant sentence in the manuscript to make it clear that we refer to continuous rotations.

Figure 2: Partly adapted from FIG.4 in the manuscript. C4-symmetry breaking in the LDOS for $\theta=0$, $\pi/2$ and π for the CSDW, MS, and SVC phase. If a continuous rotation is not experimentally possible, knowing $\delta_{C_2} = \text{Sum}_i |N(i, \theta, \omega) - N(R\{i\}, \theta, \omega)| / (N(i, \theta, \omega) + N(R\{i\}, \theta, \omega))$, where ‘R’ denotes a 90 degree rotation, for $\theta=\pi/2$ and $\theta=0$ is still enough to deduce the true ground state as seen by the distinct values at these angles by comparison of the three lower panels above.

Reviewer #2 (Remarks to the Author):

The referee writes:

“Gastiasoro et al focus on a challenging and fundamental problem of condensed matter physics, identifying a Fermi surface (FS) generated, multi-Q, spin density wave (SDW).

When a SDW is generated by FS nesting, there are multiple wavevectors (Q) related by discrete symmetries of the electronic band structure. It has proven extremely difficult to discriminate single-Q multi-domain from multi-Q single-domain SDW states, and this is especially important in compounds where iron-based high temperature superconductivity appears. Gastiasoro et al introduce an ingenious new local probe approach to solve this problem. It is based on use of field dependent scanning tunneling spectroscopy (STS) plus control of the quantization axis of the spin state of an impurity atom at the origin, Mn being the discussed. The Hamiltonian and band structure used is a model of iron-

based superconducting compounds and, in that sense, is specialized but highly relevant. The impurity Hamiltonian (Eqn. 3) is familiar.

Now, the crux of this proposal is to experimentally manipulate the impurity spin quantization axis in real-space relative to the spin density modulations of the SDW, by using reorientable magnetic field. For various SDW models the authors establish the resulting perturbation of site-specific magnetization (compared to the background order with no field) nearby; this is quite impressive and convincing in Fig 3a,b,c,d. If one could directly detect the site specific magnetization here, this problem would be solved in general. So to this point Gastiasoro et al introduce a fascinating and excellent new proposal.

The next step is to link perturbation of site-specific magnetization to perturbation of site-specific local density of states LDOS(r). Unfortunately, at least as far I can understand it, the link to experiment breaks down here. Authors state “The total LDOS measures the spectral composition of the charge density, which is coupled by symmetry only to the magnetization density squared. Therefore, any change in amplitude of the spin density will have an impact on the LDOS” etc. While true in principle, for most energies spanning all the bands of a material their amplitude of this perturbation is infinitesimally small. Such effects only become significant in some narrow energy range of coupling J_K and only for specific delocalized states that are coupled strongly to the impurity via Eqn 3.

At a practical level this is key because what is measurable as LDOS $N(r,E)$ is $dI/dV(r,E)$ where dI/dV is the differential tunnel conductance of STM tip to sample at a given site and bias voltage $V=E/e$, and this is unavoidably an energy resolved quantity. But to best of my understanding Eqn 4 and 6 are not energy resolved (or if they are this is not made clear & specific). Instead they appear to span whole band(s) thus being the site specific charge & no such measurement of site-specific net charge has even been reported.

What is measurable is the energy-resolved and site-resolved version of Eqn 6 and Fig 4a,c,e,g. For example, an energy resolved version of Eqn 6, where site-specific C4 breaking perturbations are only due to intra-unit-cell charge redistributions in a charge ordered state, have been reported in e.g. Nature 466, 374 (2010); Science 333, 426 (2011); J. Phys. Soc. Jpn. 81, 011005 (2012); New J. Phys. 14 053017 (2012); Proc. Nat. Acad. Sci. 111, E3026 (2014) and Nature Physics 12, 150 (2016). So, for this elegant and ingenious proposal to have strongest impact on the field, it is the site-specific energy-resolved $N(r,E)$ and the resulting energy-resolved Eqn 6 surrounding the reorientable magnetic impurity that is key. Given a version of Eqn. 6 and Fig 4 which identifies for experimenter how to determine the correct energies (measured from chemical potential) where the beautiful effects shown in Fig 4a,c,e,g and evaluated in an elegant new fashion in Fig 4b,d,f,h have their maximum intensity, Gastiasoro et al could then become profoundly impactful.

With such adjustments and/or clarifications, I would be enthusiastic to recommend this manuscript for publication.”

Answer:

We thank the referee for insightful comments. The LDOS defined in eq.(4) is actually energy (ω) resolved, and so is eq. (6). We apologize for the confusion and have changed notation so that this is explicitly evident in the revised manuscript, and not just in the SM. All the presented results are calculated at the Fermi level which is easily accessible by STM, i.e. we have set throughout our work $\omega=0$, but only specified this in the SM in the originally submitted version. We have now made the energy dependence explicit in eq. (4) by changing $N(i,\theta)$ to $N(i,\omega,\theta)$ and specified $\omega=0$ in the main text.

Regarding the determination of the correct energies where one should look for impurity fingerprints, one may use a simple Green's function approach to show they are not restricted to a small energy region. We start with a weak J_{KS} moment for simplicity. The full LDOS of the system for a single-impurity problem is given in the Born approximation by (quantities are matrices in orbital space, and total LDOS contains Trace over orbitals)

$$N(\mathbf{i}, \omega) = N^{(0)}(\mathbf{i}, \omega) - \frac{1}{\pi} J_{KS} \sum_{\sigma} \text{Im}[\sigma G_{\sigma}^{(0)}(\mathbf{i}, \omega) G_{\sigma}^{(0)}(-\mathbf{i}, \omega)]$$

The first term describes the unperturbed LDOS, and the second term accounts for new features in the LDOS induced by the impurity. This last term is determined by both the impurity strength J_{KS} and the real space and energy dependence of the unperturbed Green's function $G_{\sigma}^{(0)}(i, \omega)$ of the SDW state. The induced features are thus in principle present at all energies, and which energies will show the most prominent feature depend on the details of the unperturbed SDW state, which is again related to the details of the original band structure and the magnetic order parameter. We illustrate this point in Fig. 3 for a $J_{KS}=100$ meV magnetic impurity in the SVC state with $\theta=\pi/2$, and the resulting LDOS for different energy cuts at vastly different energies: $\omega \sim J_{KS}$ and $\omega \sim 10 J_{KS}$. All energies show the C2 symmetry breaking induced by the single-moment. Note also that the C2 symmetry is not localized to particular sites. Figure 4 shows the same qualitative results as in Fig. 3 but for a smaller impurity moment of strength $J_{KS}=10$ meV, and correspondingly smaller amplitude of the LDOS modulations.

Finally, going beyond the Born approximation by summing all orders for strong moments J_{KS} does not change this conclusion.

Fig. 3 Calculated local density of states, $LDOS(i, \omega)$, for a $J_{KS}=100$ meV magnetic impurity in the SVC state with $\theta=\pi/2$. For each case the energy ω is shown as inset in meV. The symmetry breaking is present at all the energies shown.

Fig. 4 Same as Fig. 3 for an order of magnitude smaller moment $J_{KS}=10\text{meV}$. Same conclusions as above but the scale is now reduced due to the weaker impurity moment.

Fig. 5: Energy dependence of δ_{C_2} (log scale) for two different magnetic impurity strengths specified in the inset.

In Fig. 5 we show the energy dependence of the parameter δ_{C_2} defined in the main text (see also caption in Fig. 2 above) as a measure of C_4 symmetry breaking. We have not managed to derive a simple formula that describes the shape of this curve for a general band and SDW order parameter, but as seen from Fig. 5, although δ_{C_2} in the present case peaks near $\omega=0$, the symmetry breaking is evident at all shown energies, and our (theory) conclusions are not valid only at limited energy ranges. Of course our description is simple in the sense that it does not include self-energy effects from interactions to electrons and bosonic modes, and we do not have inelastic processes included in the LDOS calculation. Such effects will presumably smear and limit the detection at sizable energies in real experiments, which, however, is a study that goes well beyond the current paper where all quantities are shown at $\omega=0$ in the main text.

Reviewer #3 (Remarks to the Author):

The referee writes:

“In this manuscript authors propose a new method in order to distinguish multi-Q magnetic phases in itinerant magnetic systems. The idea is based on the different polarization effects of magnetic impurities on the moments of neighboring sites which depends on the underlying magnetic structure and on the orientation of the impurity moment. Authors have in mind iron-based superconductors with a small number of magnetic impurities, e.g. Mn ions, replacing Fe. Experimentally, the holes from the Mn impurities do not delocalize and only act as local moments which couple to the itinerant magnetic

system via a Kondo-like exchange coupling J_K .

The proposal builds on the assumption that

- (a) the external magnetic field does not change the magnetic structure.
- (b) the local moment can be controlled by an external field, and
- (c) the impurity does not influence on the underlying magnetic structure.

Concerning (a), authors present evidence from the literature (Refs. 38,39) while (b) and (c) require J_K to be 'sufficiently small' which seems to be supported by the ESR experiments of Ref. 37. On the other hand, induced magnetic moments M_{ind} then are also small and even more the LDOS variations which depend on M_{ind}^2 . In this regard I don't agree with the statement on page 3 that the value of the magnetic impurity moment $J_K S$ "is not important". It matters for the expected contrast which in Fig. 3 is only labeled as 'low' and 'high'. Experimentally, the detection of these small variations may be complicated by the fact that the 'natural' cleavage plane of 122 compounds is between the alkaline and As layers whereas Fe terminations are rare (Li et al., arXiv:1602.04937).

Concerning (c), authors refer to the literature.”

Answer:

We thank the referee for good feedback. We agree with the statement that the $J_K S$ matters for the expected contrast. We meant that the size $J_K S$ does not affect the qualitative behavior, but rather the overall magnitude, as shown in Figs. 3-5 above. Whether STM can indeed detect this perturbation remains to be seen. We have reformulated the relevant sentence of the manuscript to stress that the value is not of qualitative importance. We stress that we are proposing a general strategy to distinguish single-Q from multi-Q structures, and have used the current discussion of pnictides as a concrete example.

We address the issue of the natural cleavage plane in the subsequent answer.

The referee writes:

“Authors use a 5 orbital tight-binding model for the electronic structure of iron-pnictides. However, this model is in principle derived from Wannier functions which relate the field operator $\Psi(R)$ to site ' r_i ' in the tight-binding model within a certain range $|R-r_i|$ given by the extension of the Wannier orbital (Choubey et al., PRB 90, 134520 (2014)). The analysis in the present manuscript sensitively depends on the LDOS variations on sites close to the impurity which can be significantly smeared or altered by including the structure of Wannier orbitals and therefore spoil the predicted angular dependence of the C_2 anisotropy parameter.”

Answer:

The referee raises several excellent points related to the detectability on actual real cleaved surfaces. It was not our goal to perform ab-initio material-specific calculations of the dI/dV characteristics for iron pnictides but rather to present the symmetry-related arguments how multiple-Q magnetic structure can be inferred from LDOS modulations induced by a magnetic impurity and its manipulation by an external magnetic field. We do believe, however, that for the case of pnictides our effects should be observable as discussed in the subsequent sections.

First of all, if the terminated surface is equivalent to our model lattice, i.e. Fe in the current study, then it is not important to include the Wannier functions since they preserve the symmetry of the atomic functions, and our symmetry arguments for detecting magnetic order are robust.

When the cleaved surface is NOT Fe, as is typically the case for pnictides, STM sees mainly the As atoms sticking up from the Fe-plane. In the discussion below the reference to “As ions” always refer to these “As up” lattice sites since they are important for STM. The illustration in Fig. 6(a) shows the lattice structure in the clean case. In Fig. 6(b) we show a situation with a magnetic impurity (x) substituted at the center Fe-site. Now, let us take a case where the resulting LDOS breaks C4 symmetry in the Fe plane (as discussed in the main part of our paper), illustrated by small green and yellow symbols at the NN sites to the impurity at x. In this situation the two closest As ions are equivalent, but the four As ions further away (see the large yellow and green symbols) will be distinct due to the distinct Fe sites underneath. Thus, the As lattice does NOT restore C4 symmetry in the case of C2 symmetric LDOS in the Fe plane. Note also that the extent of the C2 symmetry-breaking in real-space is quite long-ranged as seen from Figs. 3-4 which further aids the effects of C2-ness on the As ions further away from the impurity site.

In the case of C4 symmetric Fe-sites, the detected conductance map will also be C2 at the As sites. This is caused by the explicit breaking of the C4 symmetry of the top As lattice by the specific position of the impurity. This is the well-known dimer structure seen experimentally and studied by Choubey et al., PRB 90, 134520 (2014). Again, however, one should focus on the four NNN As ions to the impurity site (large yellow and green symbols in 6(b)), which will be equivalent, unlike the situation for the C2 symmetric Fe-LDOS as discussed above, and thus tetragonal symmetry is preserved on these sites.

Finally, as the referee points out, there are also additional local atomic changes caused by the impurity such as local modified hopping amplitudes, longer range potential effects, and local variations in the Wannier functions. Inclusion of such detailed atomic changes caused locally by the impurity goes well beyond the current study (and were also not part of Choubey et al., PRB 90, 134520 (2014)). While such effects surely exist, and complicate the interpretation of data, we find it very unlikely that they invalidate our proposal for pnictides. Specifically, it is hard to see how such effects should lead to a restoration of C4 symmetry in the As lattice (given an underlying C2 symmetric LDOS of the Fe lattice). Our resulting curves in FIG. 4 in the main part of the paper are not tied to sites only very close (NN or NNN) to the impurity site as seen from Figs. 3-4 above, where local atomic variations from the homogenous systems are expected to be strongest. In addition, recent theoretical studies neglecting local changes to the Wannier states find reasonable agreement to the measured conductance of LiFeAs (S. Chi et al, ArXiv:1607.03192). Lastly, let us reiterate that these are issues specifically related to pnictides; our suggestion is not tied to these systems.

Fig. 6: (a) Fe-As lattice showing the up/down As ions by filled/open symbols. (b) The same lattice as in (a) but in the presence of an impurity at the center Fe site marked by 'x'. We show a case where the Fe-lattice exhibits inequivalent NN Fe sites, i.e. a C2 symmetric situation, as indicated by the small filled yellow and green symbols. This inequality “spills over” to the NN As sites as well, as indicated by the large filled yellow and green symbols.

The referee writes:

“In summary, the work by Gastiasoro is very appealing and presents an interesting approach for detecting multi-Q magnetic phases. However, based on the arguments given above I think that the present stage of the theory is on a very simplified model level and not yet as convincing as required for publication in Nature Communications.”

Answer:

Though our modelling is simplified in the sense that it is qualitative and symmetry based, we respectfully disagree that the theory is “not yet as convincing as required for publication Nature Communications”. We propose a general new setup for using local STM to identify the magnetic ground state in any system with multiple equivalent nesting vectors. Most likely there will be systems where local impurity effects or unfavorable surface properties hampers the use of this technique, but there is not clear reason why this should generally be the case, and we have explicitly argued that for the case of pnictides, we indeed expect the main conclusions of our results to remain true when considering the actual terminated surfaces.

Reviewer #1 (Remarks to the Author):

The referee writes:

“The replies to the reviewers' comments reasonably address most of the issues, even though the authors made little change in the main text. I would recommend the publication of this manuscript if the following minor issues would be addressed.

1. I appreciate that the authors are proposing a new framework for solving the single-Q vs. multi-Q problem. Nevertheless, because a particular symmetry is broken in each case, conventional STM/STS technique might also be able to discern different ground states. Therefore, the authors should show at least one example where the conventional technique may not work well; otherwise alternative technique might not be necessary. In the reply, the authors present a couple of good examples (cuprates and pnictides) in which CDW order is hard to be detected by conventional STM/STS. I recommend the authors to describe these examples in the main text to make clear the potential of the authors' method.

2. As for the magnetic-field rotation, from experimental point of view, it is quite easy to reverse the field direction (0 and π) but it is challenging to switch the field direction from 0 to $\pi/2$ because two independent magnets should be installed in the system, or the STM should be rotatable. Nevertheless, these techniques are not impossible. If there are two magnets, a magnetic field can be applied along arbitrary direction by controlling the field strengths of the two magnets. (The highest field is limited to a few tesla.) If the STM is rotatable, field direction control is easy, of course. Thus continuous rotation is as easy as "0 and $\pi/2$ " and there is no particular advantage in the latter. I recommend the authors just to write that the field rotation is not easy but possible.”

Answer:

1. We appreciate the comment but strongly prefer not to discuss in the main text the potential observability of CDW order in various contexts. If such an order is indeed observably in e.g. the SCO phase then our proposed experiment is secondary to such a direct determination. We have mentioned why the associated CDW order may be hard to detect (its small and coincident with a Bragg position, see more below) but only the actual experiment can fully determine that. The main point is that we propose a more general way to distinguish directly single-Q from multiple-Q phases without relying of any concomitant (slaved) CDW order. Therefore, we do not wish to discuss this point in greater depth in the main part of the manuscript, where we focus on the magnetic structures.

2. This is a good point; a continuous rotation of the field is as easy and 0 and $\pi/2$. We have chosen to simply removed the small section (lines 371-378) discussing the possibility to only measure the LDOS for angles 0 and $\pi/2$.

Reviewer #3 (Remarks to the Author):

The referee writes:

“Authors have answered in detail to all points raised by the referees and have also in some parts modified the manuscript accordingly. My basic assessment of the work is still ambivalent. On the one hand I really appreciate the interesting proposal for distinguishing single- and multi-Q magnetic states which is new and definitely deserves publication in some form. On the other hand, from the results presented in the manuscript it is still not clear to me if the method can be experimentally realized. This realization requires a coupling J_K of the impurity to the itinerant moments which is small enough that

magnetic fields can "unlock the impurity moment from the itinerant magnetic configuration" (lines 170,171). However, the induced LDOS variations are proportional to the squared induced magnetization and thus also rather small. In the manuscript, authors report a value of $J_K \approx 0.8$ meV (line 168) but perform calculations with $J_K S=100$ meV (and 10 meV in the reply to referee #2). What would be the contrast for the realistic, much smaller value?

Answer:

Whether the proposed method to distinguish single- from multiple-Q phases can be realized or not, can only be ultimately determined by experiment. It will probably depend on the particular material and the particular magnetic impurity. We believe, however, that it appears likely that in the case of pnictides it could be possible and useful, and perhaps for future materials it will also come in handy to do this kind of experiment as well.

The referee asks concretely what is the contrast for realistic smaller J_K . If by maximum "contrast" one defines the maximum relative difference between the LDOS separated by a 90 degree rotation, then this quantity is obviously bias and site dependent. We have checked the results and found that for 10 meV it is $\sim 1.8\%$ and for 1meV it is $\sim 0.4\%$ of relative difference in a realistic CSDW state with a moment comparable to the experimental case ($m=0.2 m_B$). We would also like to stress that the statement "the induced LDOS variations are proportional to the squared induced magnetization" is not strictly true in this case (we have reformulated the discussion accordingly in the revised manuscript). In a metallic SDW phase, a magnetic moment directly induces a charge modulation which is proportional to the difference between G_{up}^0 and G_{down}^0 , i.e. to the magnetic order parameter (squared):

$$N(\mathbf{i}, \omega) = N^{(0)}(\mathbf{i}, \omega) - \frac{1}{\pi} J_K S \sum_{\sigma} \text{Im}[\sigma G_{\sigma}^{(0)}(\mathbf{i}, \omega) G_{\sigma}^{(0)}(-\mathbf{i}, \omega)]$$

Now, the associated LDOS modulation (which integrates to the charge modulation) is energy dependent and the maximum "contrast" can be substantially larger than the small charge modulation would suggest. This issue was discussed at length with referee #2 in the previous round of reports.

The referee writes further:

"In this context referee #1 raised the interesting point that in principle a nonmagnetic impurity (which couples directly to the density) could also distinguish the different phases. Authors show the resulting LDOS patterns for the three phases and two impurity sites in Fig. 1 of their reply. In fact, this allows to distinguish MS (C2-symmetry) from SVC and CSDW (C4-symmetry) where the latter phases can in addition be distinguished due to the inequivalent sites in the CSDW phase. Now authors argue that the corresponding contrast is small because it depends on the CDW amplitude differences. Is it really smaller than the contrast obtained with the magnetic impurity and a coupling of $J_K \approx 0.8$ meV? This lacking of an estimate for the expected contrast in an experimental realization is my only concern. The principle proposed in the manuscript is definitely new and interesting and may nevertheless challenge colleagues in the field to perform the experiment"

Answer:

The referee is correct that in principle if the CDW order is detectable in pnictides, then one may use a nonmagnetic impurity to distinguish between MS on the one hand and CSDW and SVC on the other hand (as we discuss in the SM). The presence of the CDW could then be used to distinguish CSDW from SVC. The point of our proposal is that it is a general way to distinguish single-Q from multiple-Q phases, irrespective of whether or not some of the multiple-Q phases allow for a concomitant CDW

order. Such CDW order is slaved to the spin order, and not a universal feature of multiple- Q phases, and hence it is best not to rely on this effect in general. We emphasize again that the CSDW charge order is coincident with the real underlying crystal structure and the associated small charge order peak exists on top of a lattice Bragg peak, which makes its detection notoriously difficult. Therefore, we prefer not to focus on it in the manuscript. Concretely the LDOS modulations in the CSDW phase for a magnetic order of $m \sim 0.2 \mu_B$ are of the order of 0.2%, a number which is somewhat bias-dependent and scales with the magnetic order parameter of the CSDW phase. In comparison we have checked that the relative LDOS modulations for a $J_K = 1 \text{ meV}$ magnetic impurity is of the order of 0.4% at low energies. Thus even the small 1 meV case can actually lead to larger LDOS contrast than the underlying CDW order. In the paper we have used a larger J_K for clarity, but it now says also in the paper that the optimal magnetic impurity is one that has a large coupling to the itinerant electrons (and hence at large LDOS contrast), yet still weak enough that it can be rotated with the external magnetic field. We thank the referee for making us highlight this point.